# Physical activity status among adolescents in Southern Ethiopia: A mixed methods study

Eshetu Andarge Zeleke[1,2], Teshale Fikadu[1,3], Muluken Bekele[1], Negussie Boti Sidamo[1], Kidus Temesgen Worsa[1]*

1 School of Public Health, College of Medicine and Health Science, Arba Minch University, Arba Minch, Ethiopia, 2 Flinders Health and Medical Research Institute, Discipline of Population Health, College of Medicine and Public Health, Flinders University, Adelaide, South Australia, Australia, 3 Department of Nutrition and Dietetics, Faculty of Public Health, Institute of Health, Jimma University, Jimma, Ethiopia

* kidusteme05@gmail.com

**Data Availability Statement:** All relevant data are within the manuscript and its Supporting Information files.

## Abstract

### Background

Adolescents physical activity is associated with current and future health benefits, reduction of cardio-vascular risk factors, improved bone mineral density, and mental health. The aim of the current study is to assess physical activity status and its factors among adolescents in Arba Minch and Jinka towns, Southern Ethiopia.

### Methods

The study was conducted on 1255 randomly selected schools adolescents of Arba Minch and Jinka town by employing a mixed method. The qualitative data was obtained by Focus Group Discussion. Multiple linear regressions were done to identify factors affecting physical activity. Codes, sub-categories, and main categories were derived from the transcripts and presented in narrative ways to describe adolescent student's perception on physical activity, its barriers and facilitators by comparing with quantitative findings.

### Results

The mean physical activity level was 2.08 (95% CI: 2.04–2.12). A student's self-perception about being physically active, being a member of a sport or fitness team, and engaging in after-school activity to earn money, being older, sex, a self-perception of being healthy, higher levels of vegetable and fruit consumption, having someone who encouraged physical activity, perceiving one's family as being active, self-perception of not being overweight and attending schools that have a sports/playground were factors associated with physical activity. The qualitative finding showed a related finding. Poor awareness on the recommended physical activity, benefits of physical activity, lack of interest, restrictions from family members, peers and the community, uncomfortable environment were barriers to physical activity.

### Conclusion and recommendation

The physical activity level of adolescents was low. Age, sex, a positive self-perception about PA and general health, and perception about one's family PA, healthy eating practice, and

**Funding:** The authors received no specific funding for this work.

**Competing interests:** The authors have declared that no competing interests exist.

**Abbreviations:** CDC, Center for Disease Control and Prevention; PA, physical activity; SDGS, Sustainable Development Goals; SES, Socio-economic Status; WHO, World Health Organization.

the presence of role model were associated factors. Lack of self-motivation, interest and family restrictions were barriers to physical activity. PA promotion should be made by incorporating PA into school health programs and strengthening the existing school curriculum.

## Background

The Center for Disease Control and Prevention (CDC) defines physical activity (PA) in a different way from exercise as -any bodily movement that results in energy expenditure (e.g., walking, taking the stairs) while exercise is any PA that is planned, structured, and repetitive, for the purpose of improving or maintaining one or more components of fitness [1–3]. According to report by the World Health Organization (WHO), physical inactivity has been identified as the fourth leading risk factor for global mortality, resulting in an estimated 3.2 million deaths each year [4–6]. PA is a key modifiable risk factor and a cornerstone preventive strategy for reducing NCDs. Any intervention that promotes and creates a conducive environment for PA results in reduction of health care costs, increment in economic productivity and return on investment [6]. The high magnitude of obesity and other serious medical conditions among children and adolescents that is being seen in the contemporary Europe and elsewhere in the world is majorly associated with physical inactivity during in the early years of life. Because of the increased attention to obesity in the political, media and scientific context since the late 1990's, PA has been placed as one of the current global public health priority issues [7].

A range of activities including play, household chores, recreational activities, means of transportation, planned exercises like physical education and games that children and young people engage into are PA. Evidence support that, PA provides fundamental health benefits that ranges from musculoskeletal health to cardiovascular health, neuromuscular coordination, and maintenance of healthy body weight to children and youth in the age group of 5–17. In addition to its physical health benefit, PA has been associated with psychological benefits in young people by reducing symptoms of anxiety and depression, enhancing social development through self-expression, building self-confidence, social interaction and integration [8]. Apart from its physical and psychological health benefits, a PA habit adopted during childhood and adolescence is of importance for it can remain throughout their lives [9].

Current recommendations state that young people should accumulate at least 60 minutes of moderate-to-vigorous intensity PA (MVPA) on all or most days of the week [7]. However, a review of evidence from 39 countries across the globe indicated that only few of adolescents adhere to this recommendation though high proportion of older adolescents and females are physically inactive compared with their counterparts. In the review,77% of adolescents aged 11 years (boys 72%, girls 81%) and 85% aged 15 years (boys 81%, girls 90%) were engaging in fewer than 60 min of daily MVPA [9]. Pocket studies from different countries across the world have also shown a lower prevalence of physically active adolescents or a lower compliance to the recommendations among school adolescents though variations exist among literatures in the measurement of PA [3, 10–13]. About half of American people have one or more preventable chronic diseases. Seven of the ten most common chronic diseases were favorably influenced by regular PA. Yet nearly 80 percent of them were not meeting recommendations in the key guidelines for both aerobic and muscle-strengthening activity, while only about half meet the recommendations in the key guidelines for aerobic PA. This physical inactivity is linked to approximately $117 billion health care costs annually and about 10 percent of premature mortality [14].

One of the focus areas in the Sustainable development goals (particularly goal 3) is reduction of the increasing burden of non-communicable diseases and the promotion of PA is emphasized to be one the strategies to achieve the targets. In response to this, member states of the WHO have agreed to reduce insufficient PA by 10% by 2025 [2]. As supported by studies in Sub-Saharan Africa, modern life in the era of technological advances in urban areas today is bringing greater exposure to physical inactivity and metabolic syndrome than the traditional way of life in rural areas [15–17].

In Ethiopia, it is customary to see the majority of children and young people engaging in home chores and light work every day for family help. In 2018, a global alliance for active and healthy kids (AHKGA), in its first card development in Ethiopia, estimated that 28% of children and young people (17% urban and 39% rural) meet the daily recommended 60-minute MVPA. Experts of the global initiative estimated that 71%, 48%, 13%, and 14% of children and youth actively play for at least 2 hours, walk to and from school (31% in urban and 65% in rural areas), spend on no more than 2 hours screen time (mobile phone, play station, and TV), and get encouragement and support to move from their family members (buying bike, handball, or football) respectively. An estimated 32% of schools in the country have access to infrastructure like sports fields, outdoor playgrounds, and multi-purpose spaces for PA. Approximately 8% of children and young live in environments that have the inevitable infrastructure like sidewalks to engage in PA In [18].

Based on our current understanding, there has been limited research on physical activity (PA) among both in-school and out-of-school adolescents in the country, despite the overall assessment. However, a recent study conducted by Mohammed et al. in Debre Birehan town, central Ethiopia, shed light on this topic. The findings revealed that a mere 17.2% (with a range of 14.13% to 20.27%) of school adolescents were engaged in physical activity. It is important to note that this measurement of physical activity considered one hour of moderate-to-vigorous physical activity (MVPA) on only three days within the previous week [16]. In contrast to the World Health Organization's recommendation of engaging in at least 60 minutes of moderate-to-vigorous physical activity (MVPA) on most or all days of the week, a different study conducted on the same population took a closer look at sedentary behavior. In this study, sedentary behavior was defined as spending more than 2 hours per day on screen time. The findings showed that 65.5% (with a 95% confidence interval of 61.32% to 69.08%) of students exhibited sedentary behavior. Furthermore, the study discovered associations between sedentary behavior and factors such as higher maternal education, increased access to television and the internet, and greater use of social media [17]. A population-based study in Addis Ababa recently conducted in the capital city, Addis Ababa found a gender difference in PA where higher proportion of male were engaged in MVPA than females [18]. The use of Andarge et al.'s modified PAQ-A in the current study [19] will add upon this evidence by providing wide information on the PA status of adolescents over a range of different times of a day, on each of the days in a week and also the WHO recommendation of accumulating at least 60 minutes of MVPA on all or most days in a week.

The Active Healthy Kids Global Alliance (AHKGA) was established to reverse the global burden of the pandemic of physical inactivity and childhood obesity and empower countries to take evidence-based actions. Currently the initiative, in its 1st Ethiopian card development, calls for laying a ground basis for PA and sedentary behavior research and advocate among children and youth in Ethiopia. Findings from the report card indicated a huge visible policy, practice, and research gaps on PA among children and youth in Ethiopia. In the review, majority of children and youth take part in home chores and light work every day for family help in Ethiopia. However, an estimated 28% of children and youth (17% urban & 39% rural) meet 60 minutes moderate PA every day [18].

However, to the best of our search of evidence, there is a dearth of evidence on PA levels of school adolescents and factors associated to it in Ethiopia in general and in the study area too. Therefore, this study tried to fill this gap by employing a mixed method of quantitative and qualitative studies (Convergent parallel design/concurrent triangulation) among school communities in Southern Ethiopia.

The main objectives of the current study to assess PA status and identify the associated factors among high school students in Arba Minch, and Jinka towns, Southern Ethiopia from 15 November 2020 to April, 2021.

## Methods of the study

### Study design

An institution based multi-centered mixed method was conducted by employing a mixed method of data collection (quantitative and qualitative). The data collection was conducted a year after this initial data collection from 15 November 2020 to April, 2021.

### Study setting

The study was conducted in high schools of two zonal towns (Arba Minch and Jinka) in Southern Nations, Nationalities and People's' region (SNNPR), South Ethiopia. The study is was based in towns because of the great urban-rural disparities in PA, as urban dwellers are were more exposed more to sedentary behaviors as a consequence well as of the use of new technologies in Africa (TV, Movies, Smart phones and Social Medias, etc.) [15–17]. The two towns were main towns of Gamo (Arba Minch) and South Omo (Jinka) located 454 and 691 KMs from Addis Ababa, the capital city of Ethiopia respectively. In Arba Minch, there were 9 high schools, of which 5 were governmental schools and 4 were private schools. A total of 6 high schools were there in Jinka town, four of which were governmental [9th -12th] and 2 are private schools. These 15 high schools had a total enrolment of 12,012 students in grades 10–12 at the time this study was conducted [20, 21].

### Participants

Source population of the present study were all high school students (14–19) enrolled in the high schools of Arba Minch and Jinka towns and study population were randomly selected high school students from the two towns. Inclusion criteria were all randomly selected students (14–19) enrolled in the high schools of the two towns was included in the study. Those who were not willing to participate in the study and not gave consent would be excluded from the study.

### Sample size determination

For quantitative study the sample size was calculated using Open-epi software by following assumptions for two population proportions from previous similar studies. The largest sample size (n = 1286) was obtained from a study done in Brazil with considerations of 95% confidence level, 80% power, P1 (proportion of insufficient PA among adolescents with screen time of less than or equal to 3 hours/day = 63%%), and P2 (proportion of insufficient PA among adolescents with screen time of greater than 3 hours/ = 70.8% [12]. After adding 5% of non-response rate, the total sample size was found to be 1351. For the qualitative data, six focus group discussions (FGDs) (the number of FGDs was determined based on idea saturation) were formed in groups of male and female students. Each of the groups consisted of 8 to 12 members.

## Sampling procedure

Quantitative study, high schools in Arba Minch and Jinka were stratified as government and private high schools. There were 5 government high schools and 4 private high schools in Arba Minch town. Three out of the 5 government high schools (Arba Minch Secondary School, Chamo Secondary School, and Limat Secondary School) and 1 out of three private high schools (Future Hope) were selected by simple random sampling. In Jinka town, there were 4 government and 2 private high schools. Two out of the 4 high school government high schools (Jinka Secondary School and Jinka Maremia Secondary School) and 1 of the two private high schools (Biruh Tesfa High School) were also selected using a simple random sampling. The total number of students enrolled in grades 10–12 in the seven high schools selected for this study was equal to 6,862. With the assumption of similar PA across grades and classes, this study used the total number of students per school to select the number of students required from each school with males and females in separate proportions.

For the qualitative study, FGD discussants were selected purposively by the research team together with school principals and class monitors. Thus, key discussants believed to give the necessary information on perceptions and practices of male and female adolescents towards PA and its barriers and facilitators were selected for each FGD.

## Recruitment of participants

Quantitative study: students who fulfill the sampling criteria were recruited in to the study on the day before data collection. The participants who were randomly selected from school rosters were invited by the research team using a verbal script which describes the objectives of the study, significance of the study and that participation in to the study was voluntary. Those who agreed were told the time and place of data collection. On the date data collection students were invited to class rooms specially arranged in advance in the school compound for the purposes of data collection.

Qualitative study: FGD participants who fulfill the purposive selection criteria were approached using verbal script and were told the place and time of FGD discussion in advance. Information sheet containing the title, purpose, objectives of the study, risks, benefits, and rights of participants was provided for each FGD participant before the commencement of data collection.

## Data collection tools and procedure

The level of PA was collected using the modified version the PA questionnaire for adolescents (PAQ-A) which was published from a data collected in the first phase of this survey [19]. The original questionnaire was valid and reliable measures of general PA levels in adolescent [22]. Questions assessing the independent factors were reviewed from related literatures on PA [11–15]. The questionnaire used in this study a structured form and self-administered. It was initially prepared in English then translated to Amharic by a team of teachers from the departments of both languages. Eight data collection facilitators were recruited to facilitate the data collection. Weight and Height of students was measured at exit from class after filling the questionnaire. One day long training was given to the data collectors on body mass index and the data collection facilitators for the self-administered questionnaire.

For the qualitative data, unstructured open ended and non-directive FGD guide was designed in order to triangulate responses obtained from the quantitative survey. The research team moderated the discussion among male and female discussants in the presence of note takers and technical assistants. A three days training and practical demonstration was given for the data collectors. Group discussions with the respective discussants were conducted in a separate

quite class in the school compound. Each discussion was tape-recorded in order not to miss all the issues discussed. The recordings were transcribed initially in the original language (Amharic) by experienced transcribers and later translated in to English by the research team.

## Variables of the study

Dependent variable of the study was PA status (a composite index quantitatively rated from 1 to 5 with 1 indicating the least PA level and 5 indicating the highest PA level). Independent variables were socio demographic individual variables, school environment related variables, family related variables, and social related variables.

## Measurements

PA status was measured from a self-administered 7-day recall instrument previously developed and validated by WHO [22] to assess general levels of PA for senior secondary school students between the ages of 14 and 19 years old. The questionnaire had 9 questions out of which eight questions were used to calculate activity level of the students. These study used the modified version of the scale that was published from the first phase data of these research project with 7 items for PA scoring [22].

Scoring of the modified PAQ-A involved 2 steps. First, obtaining the mean of item 8 (the item asking participants to indicate the total amount of time they were very active in doing moderate to vigorous activities). This was done by adding responses on the duration of PA on each of the seven days of the previous week and dividing the total by seven using the compute command under transform menu in the SPSS. Second, the average of items 2 to 8 was computed. Initially, the scores given to items 2 to 7 and the mean of item 8 were added. The sum of this was divided for the number of items (7) to obtain the mean PA for each student (the average PA scores of each student) which falls in the range of 1 (very low PA level) to 5 (very high PA level). This score was treated in its continuous form as the dependent or outcome variable to build a linear regression model. Wealth index (WI) was constructed by considering 15 variables that were prospectively indicative of wealth. Principal Components Analysis (PCA) were done after checking the fulfillment of assumptions and factor score was calculated and the scores were subsequently ranked and categorized into quintiles. Body Mass Index (BMI) was calculated. Weight of the adolescents was measured to the nearest 0.1 kg using calibrated portable electronic digital scale (Seca, Germany model) and height was measured to the nearest 0.1 cm using a portable height measuring board following standard anthropometric techniques.

## Bias

Even though the questionnaires were filled in self-administered bases, there is potential of social desirability bias when facilitators move around in the class room were the data was collected. During the training a through training was provided to the facilitators of data collection so that they. Students were given instruction to fill in their answers to physical activity questions while the facilitators read each questions being at the stage of the classroom. Moreover they were also told that the questionnaire was anonymous only the codes were available on the questionnaire sheet.

Since students of the same family or neighbors might have been enrolled in same schools, there might be a risk of information sharing among the participants. To reduce this, data were collected at the same time from different schools included in to the study from the same town.

## Statistical methods

For the quantitative data, the data was entered into Epi-data software version 3.1 and then exported to SPSS version 25 statistical package for analysis. Descriptive statistical analysis was done and summarized by tables. In order to identify the variables that were predictive of PA scores, a series of standard (simultaneous) multiple regression analyses was conducted. For anthropometric data analysis, standard deviation (z-score) scores were obtained by WHO Anthro plus software to determine the nutritional status of the student. Accordingly, overweight = BMI for age>1SD (equivalent to BMI 25kg/m2), obese = BMI for age>2SD (equivalent to BMI 30kg/m2), thin = BMI for age< -2 to -3SD, severely thin = BMI for age <-3SD and stunted = Height for age <-2SD to-3SD. The qualitative data was analyzed using open-code software version 4.02. The transcribed document was exported to the software and codes, sub-categories; main categories (themes) were formed using the principles of conventional inductive qualitative content analysis. Findings were presented in narrative ways by comparing and contrasting with quantitative findings.

## Data quality assurance

To assure the quality of data, thorough training was provided to the data. The collected data was reviewed and checked for completeness before data entry. The variables were coded, and then edited during data entry. For the qualitative data, purposive selection of informative discussants was made by involving school officials and class monitors that know the background of their students.

## Ethical consideration

The study was conducted according to the guidelines of the Declaration of Helsinki, and ethics approval was obtained from Arba Minch University's College of Medicine and Health Science, Institutional Ethics Review Board (Reference number IRB/116/11). Prospective participants were informed about the purpose of the study, that participation was entirely voluntary and anonymous, and that they could decline the invitation to participate without any negative consequences. An informed written consent was obtained from all participants of the study. Those participants aged 16 and above signed off on a written consent sheet provided to them before filling in the questionnaire. An additional written consent was obtained from school administrators and parents, or guardians of minor participants aged below 16 years. The questionnaire has not contained names of the study participants apart from codes assigned to each questionnaire. However, the research team had access to other identifiable information like age and school which will be kept confidential to the research team.

## Results

### Socio demographic and personal characteristics of the study participants

Out of 1351 adolescents, 1255 participated in the current study, yielding a response rate of 92.9%. Of the non-responses, 72 filled incomplete information and the rest refused to participate in the study. The remaining was incomplete. Thus, 32%, 31.6%, and 36.5% of students participated from 10th, 11th and 12th grade students. Grade 9th students were not available due to school closure. More than half of the participants (52.3%) were females. The median habitual time of sitting for study, watching TV, and playing games combined was 120 minutes (minimum 35 and maximum 450 minutes) whereas the median was 80 minutes (minimum 30 and maximum 360 minutes) for watching TV and playing games combined. There was an approximately comparable habitual time of sitting for study after school; 45 minutes (SD = 20

minutes, maximum = 90 minutes), watching television; 50 minutes (SD = 40 minutes, maximum = 300 minutes), and playing games; 50 minutes (SD = 40 minutes, maximum = 240 minutes). The mean minutes spent watching television and playing games added together was 100 minutes (SD = 60 minutes, maximum = 360 minutes).

Regarding their perception of themselves, 7.1%, 13.6%, and 35% perceived their general health, PA, and body weight as not good, less active than others, and an abnormal weight respectively. Considering every use of either of the substances (alcohol, khat, and cigarettes), 264(21.0%) reported that they had used those substances. The details of other characteristics of the participants are indicated in Table 1.

Regarding their perception about PA, its benefits and the outcomes physical inactivity, boys and girls discussants have a different feeling and understanding. Boys were interested in doing a PA though some argued that it makes them to get tired and affects their study time. Most boy students knew that a good PA prevents from developing diseases like heart diseases, hypertension and diabetes. In practice, however, there was a great variation among the discussants. A 19 years old boy from a private school said "*I am too much interested in doing a PA since my elder brother and my father encourage me to be both strong and clever student. My father advises me that PA is important for making the brain active, and he helps me to get up early in the morning and to jog along the street or to lift a weight of various kilograms in our own compound*". On the contrary, most participants from two female FGDs from private schools indicated that they do not have an interest in doing a strenuous PA as it makes them get tired, and makes them go out and become rude. In the male FGD, some students argued PA as a barrier affecting their study time, and hence not interested to get committed to it. A 16 year female discussant said "*I do not generally want to perform a PA because I easily feel tired, it might also deter me away from my studies, and I am sorry, I do not believe in its benefits in preventing disease too. There are so many elders who stay on bed for more than 20 years because of a muscular or nerve problem, and who do not die of the dangers diseases like hypertension, diabetes, heart failure, and etc.*"

### School environment and peers related factors

Nine hundred ninety one (79%) participants reported that they attended physical education at school. The remaining 264(21.0%) did not yet start PE class. Most of the participants reported that they enjoyed PE class or any PA done out of the PE class both in-and-out of the school. Five hundred sixty-seven (45.2%) of the students reported that there was someone who encouraged them to engage in PA, and nearly half (48.5%) of them were peers in the school. More than one-third (38.6%) of the students reported that there was a khat chewing or game playing shop around their school environment (**Table 2**).

### Household and family characteristics of the students

Approximately 1 in 5 of the students' parents had not received any formal education, but a larger proportion of the students' fathers (1 in 3) were educated beyond grade 12 whereas only 1 in 5 mothers were educated so. Regarding parent's occupation, approximately 40% of both parents were reported as government employees. More than half (58.1%) perceive their family as physically active, and 61.1% of the students reported as there was chronic diseases in their family (Table 3).

### PA status of students

Regarding the types of students' PA activities listed under item 1 of the modified PAQ-A, the common activity was walking quickly for exercise, 31.1% of the girls and 39.3% of the boys done on three or more occasions in the previous week. The second most common activity was

**Table 1. Individual characteristics and behaviors of high school students from Arba Minch and Jinka towns, Southern Ethiopia, 2021.**

| Variables (N = 1255) | Category | Count | Percentage |
|---|---|---|---|
| Age | Less than or equal to 15 | 46 | 3.7 |
| | 16 | 190 | 15.1 |
| | 17 | 350 | 27.9 |
| | 18 | 373 | 29.7 |
| | 19 | 296 | 23.6 |
| Grade | 10th | 401 | 32.0 |
| | 11th | 396 | 31.6 |
| | 12th | 458 | 36.5 |
| Sex | Male | 599 | 47.7 |
| | Female | 656 | 52.3 |
| BMI | Underweight and below | 66 | 5.3 |
| | Normal | 1048 | 83.5 |
| | Overweight and above | 141 | 11.2 |
| Religion | Orthodox | 660 | 52.6 |
| | Protestant | 546 | 43.5 |
| | Muslim | 49 | 3.9 |
| Marital status | Married | 12 | 1.0 |
| | Unmarried | 1240 | 98.8 |
| | Divorced | 3 | 0.2 |
| Residence type | House having compound | 1163 | 92.7 |
| | Apartment | 92 | 7.3 |
| Use transportation to school | Yes | 525 | 41.8 |
| | No | 730 | 58.2 |
| Kind of transport used | Active (bicycle) | 141 | 26.8 |
| | Passive (*car, motorcycle or three wheel drive taxi, etc.*) | 384 | 73.2 |
| Work after school to earn money | Yes | 648 | 52.0 |
| | No | 599 | 48.0 |
| Member sports or fitness team | Yes | 462 | 36.8 |
| | No | 793 | 63.2 |
| Self-perception of general health status | Very good | 597 | 47.6 |
| | Good | 569 | 45.3 |
| | Not good | 89 | 7.1 |
| Self-perception of PA | More active than others | 545 | 43.5 |
| | As active as others | 538 | 42.9 |
| | Less active than others | 170 | 13.6 |
| Self-perception of body weight | Normal | 812 | 65.0 |
| | Overweight/fat | 344 | 27.5 |
| | Underweight/thin | 94 | 7.5 |
| Smoke cigarettes | Yes | 151 | 12.0 |
| | No | 1104 | 88.0 |
| Drink alcohol | Yes | 133 | 10.6 |
| | No | 1122 | 89.4 |
| Chew khat | Yes | 118 | 9.4 |
| | No | 1137 | 90.6 |
| Frequency of vegetable or fruit consumption | 1–2 times or not consumed | 484 | 38.6 |
| | Consumed 3 and more times | 771 | 61.4 |
| Use of either of the drugs (*alcohol, khat or cigarette*) | Having no habit of using drugs | 991 | 79.0 |
| | Having the habit of drug use | 264 | 21.0 |

**Table 2. School environment and peer related factors among high school students in Arba Minch and Jinka towns, Southern Ethiopia, 2021.**

| Variables (N = 1255) | Category | Count | Percentage |
|---|---|---|---|
| Attend PE at school | Yes | 991 | 79.0 |
|  | No | 264 | 21.0 |
| Enjoy PE class or any PA | Yes | 1081 | 86.1 |
|  | No | 174 | 13.9 |
| School have a play ground | Yes | 878 | 70.0 |
|  | No | 377 | 30.0 |
| School provides equipment for PA (*ball, clothing, etc.*) | Yes | 520 | 41.4 |
|  | No | 735 | 58.6 |
| Presence of any person who encouraged PA | Yes | 567 | 45.2 |
|  | No | 688 | 54.8 |
| Person who initiated for PA (role model) | school peers | 275 | 48.5 |
|  | peers in the town | 222 | 39.2 |
|  | teachers | 63 | 11.1 |
|  | Others* | 7 | 1.2 |
| Presence of khat chewing or game playing areas near by the school | Yes | 484 | 38.6 |
|  | No | 771 | 61.4 |

* Parents, people from the community who initiate mass sport

different among girls and boys; 17.2%) of girls reported that they jogged or ran whereas 27.9% of boys reported that they played football. Least involvement was associated with volleyball and basketball for both girls and boys. Around 1 in 3 (33.7%, n = 202) boys reported that they were engaged in other types of PA whereas approximately 1 in 5 girls (18%, n = 118) reported that they did so. Only a few of the students reported that they had done other PA activities (n = 40). However, not all of them mentioned the activities though few reported those activities comprised attending gymnastic centers, lifting weights and home-based pushups, and other local sports.

Results from the FGD discussions showed that most of the discussants agreed that a PA is a programmed sport activity, and not any kind of movement that can be done at household level. Boy discussants inclined to in-door sport activities that build muscles as useful types of activities that can be considered as PA. These include lifting weight, doing a push-up, and jumping over a rope, etc. Some boy discussants discussed that they attend to aerobic activities regular at gymnastic centers. Female discussants admitted that they are not actively engaged in MVPA, and simply do routine household chores which can be classified as light intensity PAs. A 17 years old girl from a government-owned school said ". . . . . .*In fact, we do not have a scheduled PA like that of our boy peers, sometimes we are allowed to stretch our legs if household chores are already completed*". Female discussants also believed that they prefer walking to any other sports for they feel that it makes them to be descent, has a good support from same sex peers, and does not make them to get tired, and distracted from their studies. No discussant was able to classify PA as light intensity, moderate intensity and vigorous intensity activities. However, when they mention their activities the majority do not perform activities that can be classified under MVPA.

In their response to items 2–6 of the modified PAQ-A, only few students (1.9%) reported that they were very active quite often during their PE class. More than half (54.5%) of the students reported that they engaged in to some degree of activity from sometimes to most times during the lunch time other than sitting and eating their lunch. Regarding their activity right

**Table 3. Family-related factors among high school students from Arba Minch and Jinka towns, Southern Ethiopia, 2021.**

| Variables | Category | Count | Percentage |
|---|---|---|---|
| Father's education | No formal education | 210 | 16.73 |
| | 1–8 grade | 406 | 32.35 |
| | 9–12 grade | 267 | 21.27 |
| | Above 12 grade | 372 | 29.64 |
| Mother's education | No formal education | 279 | 22.23 |
| | 1–8 grade | 495 | 39.44 |
| | 9–12 grade | 242 | 19.28 |
| | Above 12 grade | 239 | 19.04 |
| Father's occupation | Government employee | 506 | 40.32 |
| | NGO employee | 272 | 21.67 |
| | Private worker | 365 | 29.08 |
| | Daily laborer | 94 | 7.49 |
| | Others* | 18 | 1.43 |
| Mother's occupation | Government employee | 473 | 38.05 |
| | NGO employee | 307 | 24.70 |
| | Private worker | 312 | 25.10 |
| | Daily laborer | 123 | 9.90 |
| | Others** | 28 | 2.25 |
| Perceived family's PA | Active | 729 | 58.09 |
| | Passive | 526 | 41.91 |
| Chronic disease in the family | Yes | 488 | 38.88 |
| | No | 767 | 61.12 |
| Wealth index in quintiles) | Lowest | 250 | 19.98 |
| | Low | 253 | 20.22 |
| | Middle | 217 | 17.35 |
| | High | 254 | 20.30 |
| | Highest | 277 | 22.14 |

*Receiving a pension, receiving disability assistance

**housewives, receiving a pension

after school, about a third (32.7%) of them reported that they were very active on 2 or more days in the previous week. A similar percentage of the students (31.5%) were very active during the night time (dusk time) whereas a relatively lower proportion of them (27.2%) were very active for two or more times during the week end.

Regarding responses to item 7 of the PAQ-A (general activity during free time in the previous week), 48.8% of girls reported that they had spent all or most of their free time doing little PA whereas 34.6% of boys reported in the same way. The sex difference in spending free times doing little PA in the previous week was statistically significant, $\chi^2$ (1, $N = 1,255$) = 10.63, $p = .001$. Conversely, few (6%) reported 5 for item 7 (being physically active on at least five occasions in their free time in the previous week).

Item 8 of the modified PAQ-A assesses each day's PA activity in the previous week. Of the students participated in the survey, 195 (15.5%: 21.5% of the girls, 9.0% of the boys) reported that they did not do any moderate to vigorous PA in the previous week. The vast majority (72.3%) of the 195 students were girls whereas nearly 1 in 3 were boys $\chi^2$ (1, $N = 1,255$) = 249.03, $p < .001$). Only 65(5.2%) of the students reported a mean of 4 and above on item 8 of the PAQ-A, and could therefore be regarded as meeting the most-recent WHO

recommendation of 60 minutes of MVPA on most days of the week. Of these, more than half (53.8%) were boys and 46.2% were girls $\chi^2$ (1, N = 1,255) = 7.23, *p* = .007). Few (3.2%) adhered to a daily 60 minutes and above PA in the previous week by reporting an average of 5 for every day on question 8. Among these 40 students, 17 (42.5%) were girls, 23 (57.5%) were boys; 40% were 19 years of age. Among the days, the highest level of PA (mean = 2.49 out of 5) and the widest SD (SD = 1.43) was recorded on Saturday; with other daily means ranging from 2.06 to 2.26, and the *SD*s ranging from 1.16 to 1.37.

The qualitative finding explored a low awareness of the existence of any standard on the duration, intensity, and type of PA among the FGD discussants. Almost all of them do not know that there is a recommended duration and type of PA for different population groups or in different age brackets. Some of the participants discussed that they know a 30-minute activity (irrespective of intensity) is required at least three times per week from any person. Almost all the discussants did not have the awareness that a daily average of 1 hour MVPA is recommended for their age. Those who had an awareness about any standard were aware of the recommended standard for adults (30 minutes for 3 days per week of any intensity exercise), and they considered that as a standard to their age too. Few of the discussants who were aware about this standard reported that mass media were their primary source of information. This might imply that mass media broadcasts might not be tailored in addressing the recommended PA in different age brackets. A 15 years old female student from a private school said" *our PE class majorly focuses on theoretical aspects of types of PA that we commonly practice in the fields, no one told us about the presence of a standard duration of activity for our age. I prefer the information on practical sport programmes on a TV where sometimes experts speak, in the middle of the activity, about a kind of information on duration, types of activities and their benefits to patients and healthy individuals. It seems to me an activity for luxury purpose since sport activities are done by the rich in gymnastic centers and huge resorts. However, even in those sport programmes the focus is on how to reduce weight, and prevent diseases. I do not know, I did not hear anything about standard duration among adolescents, I thought a 30 minutes PA at least 3 days per week also works to me*".

One hundred seventy eight (27.1%) and 186(31.1%) of girls and boys respectively reported that they had been engaged in less PA as compared to their previous usual PA. Among these, few (only six girls and eight boys) provided reasons for their lowered PA like being sick, menstruation, being engaged in to study during the spare time, and over-sleeping (**Table 4**).

Winsorzing was done to correct outliers and to avoid distorted results, particularly for the multiple regression. Twenty scores were winsorized with the high outlying scores defined as those exceeding 1.5 x the interquartile range within the fourth quartile. The final PA score was calculated after winsorzing the outliers.

The mean PA was 2.08 (95% CI: 2.04–2.12) out of the 5 points rating of PA in the modified PAQ-A. It is lower for girls (M = 1.95) than for boys (M = 2.22), respectively with a statistically significant on t-test for comparing means; *t*(1,253) = 7.06, *p* < .001, *d* = 0.40). The mean PA was found to be higher for students who reported that they had less PA in the previous week than their previous usual PA (*M* = 2.14) than those who reported that they did PA in the same way with their previous usual activity (*M* = 2.06). However, the difference was not significant on t-test, *t* (1,253) = 1.86, *p* = .062) (Table 5).

## Factors associated with PA

In preliminary checking, the assumption of linearity was not met by BMI, the WI, and sedentary time, so those variables were not included in the multiple regressions. In the first multiple regression, approximately half of the predictors were not significantly related to PA, so

**Table 4. PA status among high school students in Arba Minch and Jinka towns, Southern Ethiopia, 2021.**

| Variables | Category | Count | Percentage |
|---|---|---|---|
| Type of PA for three or more times in previous week Girls | Bicycling | 67 | 10.2 |
| | Jogging or running | 113 | 17.2 |
| | Aerobics | 98 | 14.9 |
| | Walking quickly for exercise | 204 | 31.1 |
| | Swimming | 103 | 15.7 |
| | Dancing | 105 | 16.0 |
| | Football | 90 | 13.7 |
| | Volleyball | 53 | 8.1 |
| | Basketball | 56 | 8.6 |
| | Other kinds of vigorous exercise* | 118 | 18.0 |
| Type of PA for three or more times in previous week Girls | Bicycling | 106 | 17.7 |
| | Jogging or running | 140 | 23.4 |
| | Aerobics | 123 | 20.5 |
| | Walking quickly for exercise | 235 | 39.3 |
| | Swimming | 127 | 21.2 |
| | Dancing | 101 | 16.9 |
| | Football | 167 | 27.9 |
| | Volleyball | 77 | 12.9 |
| | Basketball | 73 | 12.2 |
| | Other kinds of vigorous exercise* | 202 | 33.7 |
| Self-description of PA in free time during previous week Girls | Most time spent on little physical effort | 320 | 48.8 |
| | Sometimes physically very active | 227 | 34.6 |
| | Very active at least three times | 109 | 16.6 |
| Self-description of PA in free time during previous week Boys | Most time spent on little physical effort | 220 | 36.7 |
| | Sometimes physically very active | 241 | 40.2 |
| | Very active at least three times | 138 | 23.1 |

*Lifting weights, pushup, attending to gymnastic sports, etc.

backward elimination was used to successively remove predictors with the highest non-significant $p$ value(s). For the final model, in which all non-significant predictors had been removed and 11 significant predictors of PA were retained, $F(11, 1235) = 27.11$, $p < .001$, the adjusted $R^2 = .19$. According to the standardized β coefficients, a student's self-perception about being physically active was the strongest predictor of PA, followed by being a member of a sport or fitness team, and engaging in after-school activity to earn money, being older, and sex (being a boy). These five variables were followed, in order of decreasing predictive strength, by a self-perception of being healthy, higher levels of vegetable and fruit consumption, having someone

**Table 5. Summary of PA among school adolescents at Arba Minch and Jinka towns, Southern Ethiopia, 2021.**

| Metric | Girls (*n* = 656) | Boys (*n* = 599) | Customary activity (*n* = 891) | Reduced activity (*n* = 364) |
|---|---|---|---|---|
| Minimum/maximum | 1.00/3.95 | 1.00/ 3.98 | 1.00 / 3.98 | 1.00 / 3.96 |
| Mean | 1.95 | 2.22 | 2.06 | 2.14 |
| SD | 0.67 | 0.68 | 0.70 | 0.66 |
| Median | 1.85 | 2.17 | 2.00 | 2.07 |
| Skewness | 0.68 | 0.52 | 0. 66 | 0.34 |
| Kurtosis | 0.00 | −0.03 | 0.07 | −0.37 |

who encouraged PA, a self-perception of not being overweight, attending a school that did not have a sports/playground, and perceiving one's family as being active (**Table 6**).

## Barriers and facilitators of PA

Themes emerged from the qualitative study (FGDs) indicated that issues arising from the students themselves (lack of time, lack of interest or unfavorable attitude, lack of knowledge, a perception that they are not at risk), their families (lack of family support), and their environment (lack of support from the community, school environment, bad weather condition) were found to be barriers to PA among adolescents in the study area. On the other side, as facilitators, the discussants indicated family support, peer support, self-interest, perceived health benefits, advice from experts on the media, a supportive environment (home near stadium, and small field for football games, and the presence of Marshal Arts training). There were quite similar discussions from both the private and government schools though differences were there among sexes from any of private or government schools.

In summary, both male and female students do not have a good knowledge regarding PA, its benefits and a standard recommendation at their age, but male are more interested in static muscle building sport activities than aerobic MVAP recommended by the WHO while females are not generally interested in any of PA. In practice, most discussants from the male group reported that they play football, volley ball, and jog early in the morning in a mass sport in the week ends and individually on the other school days. Some female discussants reported that they do a kind of PA like jumping over a rope, and aerobic exercises at home during a practical aerobics sport programme concurrently along with different channels on a TV. However, most female participants admitted that they do not have any PA schedule except their usual work as a household chore and a subsequent walk along the road. None of the students knew that walking quickly (brisk walking) was more beneficial than any kind of routine walks they make along the street. Two typical reflections were presented from the discussants as barriers and facilitators respectively.

An 18 years old 11[th] grade female discussant from government school said "*In the first place, I personally do not have an interest to get tired of PA. That is because I do not feel that I am at risk of the diseases that PA is deemed to prevent from. I am young and I am not at risk of a*

**Table 6. Factors associated with PA among high school students from Arba Minch and Jinka towns, Southern Ethiopia, 2021.**

| Variables [a] | Unstandardized coefficients | | Standardized coefficients | | |
| --- | --- | --- | --- | --- | --- |
| | B | Standard error | β | t | p |
| Constant | −.21 | .29 | | 0.71 | .476 |
| Self-perceived PA (good) | .31 | .05 | .15 | 5.77 | < .001 |
| Member of sport or fitness team | .21 | .04 | .15 | 5.67 | < .001 |
| After-school activity to earn money | .20 | .04 | .14 | 5.48 | < .001 |
| Age (higher) | .08 | .02 | .13 | 4.99 | < .001 |
| Sex (boy) | .18 | .04 | .13 | 4.80 | < .001 |
| Self-perceived general health (good) | .27 | .07 | .10 | 3.76 | < .001 |
| Vegetable and fruit consumption | .14 | .04 | .10 | 3.89 | < .001 |
| Having someone who encouraged PA | .13 | .04 | .10 | 3.70 | < .001 |
| Self-perception of being overweight | −.13 | .04 | -.08 | -3.17 | .002 |
| Having a school with a sports/play ground | −.11 | .04 | -.08 | -2.89 | .004 |
| Family perceived to be physically active | .09 | .04 | .07 | 2.54 | .011 |

[a] Variables are listed according to size of the standardized beta coefficients.

*disease. Besides, this is the time when I have to study hard and join a university. I do not know, if I get the motive, I will have a plan to do a PA. Even if we plan to do it, both our family and the community does not support us to do; they will easily label us as a "rude". In fact, I also believe in this; a female has to do PA being at home. I only sit just to study and at the other time I do not get rest as I help my mother with the household chores. Cannot this be a PA?. laughing. . .. We also do PA during our PE class which will actually start when we finish the theory class".*

On the other side, a 16 years boy from a private school said, "*I have an elder brother having a usual programme of jogging (3 times per week) and weight lifting for the other three more days. We usually do not do any activity on Sunday for we go to the church and take rest on that particular day. I am doing a PA together with him for he advises me to do so and I have also seen its benefits. Doing a PA does not waste your time; it rather saves your time. . .you will have a good breathing capacity, and you will be healthy. For example, I have not ever been absent from school because of any disease to the extent of not acquiring a common cold. In addition to my brother, I have been advised by my friends, and I have also attended to experts' advice on the TV*".

## Discussion

This study assessed high school adolescents' PA status and the associated factors at two towns in the Southern Ethiopia using the modified PAQ-A scale published earlier [23] based on a data obtained from this project for the purpose of assessing the scale in the Ethiopian context. From the 5 points scoring of adolescents' PA, the average PA of adolescents was low as it is the case in most reports from studies which used the previous PAQ-A scale [12] as well as other studies and reviews across the world, which assessed PA of adolescents based on either the WHO recommendation or which used a different scale to measure adolescents PA [11, 13, 22, 24–26]. In some countries, a relatively better practice of PA was also reported among adolescents, 79.2% and 76% were categorized as active in another setting in Malaysia [27] and Vietnam [10] respectively.

Given PA is a strong factor in reducing the occurrence of chronic degenerative diseases and has many more benefits to adolescents [7, 9, 28–32], and adolescent age is the most appropriate age for developing habits of PA [33], the low PA among adolescents suggests that there is a need for enhancing efforts so that adolescents can meet the WHO standard of PA recommended for their age and consequently uphold a healthier generation.

Regarding the type of activities, walking quickly for exercise was the most commonly practiced activity with more than 30% of boys and girls engaged in at least three times in a week. The participation in the other activities (playing football, volley ball, swimming, etc.) was noticeably lower though boys were found to have a higher range of activity than girls (12 to 20% versus 8 to 20%). The possible reason for these differences might be the socio-cultural prejudices or barriers which limit girls from being engaged in to outdoor activities but for walking along the street with their friends. Findings from the qualitative study has also supported this since female discussants themselves believed in the culture and support that doing PA out of home compound will make them rude.

Although walking quickly for exercise (brisk walking) is recommended as a PA, it is not well practiced seen in light of the vast majority of students not being engaged in it too, and Besides, it may also be wrongly reported as an MVPA while it were the usual kind of walking along the street; walking whereas talking and playing with friends. Except for football among boys (28%), none of the forms of PA other than walking were engaged by the students to a noticeable level. The FGD discussants also stressed on few variants of PA, and that they do not know the difference between walking quickly and a routine walk meant for recreation. This

might suggest a low awareness of the benefits of the activities or an uncomfortable environment to practice those activities listed to them as can be common to Ethiopian adolescents.

The non-adherence of 95% of the students to the WHO recommendation, as reported in question 8 of our modified scale [23], is higher than a pooled figure reported by WHO (81%) in 2018 [2], and a previous study from Ethiopia (83%) [26], and other parts of the world (68%) [10, 11]. This might imply a poor knowledge about the WHO recommendations or an unfavorable attitude towards conforming to it when there is a good knowledge. A further exploration in the FGDs has explicitly shown this among both male and female discussants from both private and public schools. None of the discussants knew that an average of 60 minutes MVPA is required at their age though few were mentioning a 30 minutes PA (without any mention of its intensity) for 3 days in a week which seems closer to that WHO recommends for adults.

The most challenging issue in comparing PA of adolescents is a difference in measurement. However, irrespective of the differences in measurement, there are consistent reports about low PA among adolescents [7, 11, 13, 24–26] and higher level of PA among boys when compared with girls [7, 10, 12, 22, 27, 34]. Although not common, more boys failed to meet a satisfactory level of PA (23.8% boys versus 15.4%) in a review of PA in school children aged 13 to 15 from 34 countries where most of them were developing countries [35]. In the present study, even if there was a significant difference in PA between boys and girls, this was not the case among students with the highest PA scores. This might be because students who practice higher levels of PA might consistently maintain their practice regardless of sex differences. Sex difference was not only significant at simple relationship level using chi-square. Multiple linear regressions have also shown a significant difference among boys and girls with boys having higher PA scores than girls. This suggests that the vast majority of school adolescents who do not meet the recommended weekly PA need to be addressed in a more targeted and gender-sensitive approach apart from universal stepping to an improved PA among adolescents. However, in this study, a number of other variables were associated with students' PA. These include students' self-perception of having good PA (comparable or better PA than others), being a member of a sport or fitness team, engaging in after-school activity to earn money, and being older.

The students' self-perception of having had an equal or better PA than their peers was one of the factors associated with PA of adolescents in the current study (β = 0.15). The association is in line with a previous study from Brazil [12]. This might be explained by the presence of few students having a high PA score in the study, and those students might have background knowledge of the recommendations for performing PA, and its benefits and might be having the perception that they are doing well in relation to others in the town. On the other hand, it is likely also that those students who perform a better PA will have a good perception about their PA. Given the study is a cross-sectional survey; the chance of having a chicken-egg dilemma is expected. Differentiating between the real directions of this association is left for a future research which would require a stronger design. Self-perception of good general health and good PA among families were also positively associated factors among adolescents in the study (β = 0.10 and β = .07 respectively). This can suggest that a positive self-perception towards PA of one self and one's family, and general health might be motivating factors for a high PA practice among adolescents. However, a perception of oneself being overweight (β = -0.08) was a demotivating factor contrary to what is reported from a previous Malaysian study where a self-perception of underweight was associated with physical inactivity [25]. In the same study, BMI (being obese or underweight) was associated with a physical inactivity, and in a Vietnamese study, overweight was associated with a low level of PA. This implies the association between a perceived weight, real weight and BMI (a composite indicator of nutritional status based on one's weight and height) and PA of adolescents needs a further investigation

owing to the controversial finding from different studies. In our study, BMI did not show a linear relationship with the students PA score at the outset.

The adolescents who are a member of a sport or fitness team in the present study had higher PA scores than their counterparts (β = 0.15). Consistent with a finding reported in a previous study [10], it can be explained by the extra time these groups of students get during a scheduled PA at sport or fitness centers, and its chance of being repetitive and regular in the week as compared with self-initiated PA practices which might be interrupted or done sporadically. This suggests the need to encourage the establishment of sport or fitness centers in addition to improving awareness and attitude towards PA.

Although socio-economic status as a wealth-index did not fulfill the assumptions for multiple regression analysis, engaging in after-school activity to earn money, which can reflect either a low or high household wealth, was found to be the other factor associated with students PA (β = 0.14). This is in agreement with a finding from the Peruvian study [11]. In fact, most activities in which adolescents (particularly those from the peri-urban areas) might be engaged to earn money (travelling to nearby market places, selling lottery tickets, carrying heavy objects, etc.) can be laboring because they are not sufficiently matured to get employment at office level or may not have the capacity to have a commodity that can be bought or sold at market places. However, there would also be a chance of being more inactive while sitting in super-markets or shops as a means of helping household members during out-of-school hours. These groups of students are likely to belong to few household members with additional business shops or who are merchants, and hence might not have affected the great majority of adolescents who might have been engaged in activities that made them mobile. This finding would suggest that engaging adolescents in to a household income generation might be an indirect way of investing or accumulating into their daily PA.

Age of the students was also one of the factors associated with their PA; the older the age, the higher the PA of adolescents (β = 0.13). This is inconsistent with findings reported from previous works [12, 25, 36] and consistent with a finding from Vietnam [10]. Although cross-country and regional comparison of PA can be affected by many factors such as weather, culture, technology and access to information and health care, the high PA among older adolescents can be explained by the involvement of older adolescents in football; the commonest sport observed among out-of-school peers. Besides, older adolescents are likely to get adequate information regarding the benefits of PA and might have been positively influenced by their school peers from their experience in the schools and the mass media. However, the inconsistent finding with most previous studies needs a further investigation of the relationship between age and PA among adolescents. Another interesting, but contradictory finding relative to previous research [10] is a low PA at schools having a sport field/ playground (β = -.08). This seems an illogical association, but it might have arisen from the weakness in the questionnaire itself (PAQ-A) which, from its very nature, did not incorporate an item on a PA done during breaks in the school hours or when their teacher leaves class early or is not there. Adolescent students from a supportive school environment (offering equipment or having a sport fields) are seemingly be physically active during school hours or any time when they are in the schools. Such students, out of being tired, may not do PA at other spare times (lunch, after school or during the evening), and hence, having a reduced over all PA score.

A higher level of vegetable and fruit consumption is positively related to students PA (β = .10). This is consistent with a previous study in Brazil, Peru and South East Asia [11, 12, 37]. This might be related to familial habits where those households having the habit of consuming fruits and vegetables might be those who are informed about health promoting behaviors [11], and are consequently aware of the benefits of consuming fruits and vegetables, take care of their health, and are also likely to have the knowledge, and positive attitude towards performing PA.

The other independent factor associated with PA of adolescents was having someone who encouraged PA (peers in the school or out of the school, teachers, family members, etc.) ($\beta$ = .10). A previous study also indicated a related association with PA of adolescents where family support, intimacy and involvement in PA were important factors [38]. Adolescents who received encouragement are likely to be motivated, and perform PA in a more continuous and sustained manner than are those who could not get such kind of support. Most discussants in the FGDs also stressed the importance of support from family members, peers, and the society as a facilitator to PA by reflecting that where the support is there a good PA will also be there. This suggests that promotion of PA needs to be strengthened in a network of friendship, school and, familial environments, where investment in a given cohort may have an effect at team level.

Although the barriers and facilitators reported in the FGDs did not fit exactly to the factors associated with PA among adolescents, they are generally related in that they stemmed from individual adolescent's interests, motivation, supportive familial and environmental conditions, societal attitudes and perceptions.

## Strength and limitations of the study

Strength: The study was conducted among large number of school adolescents. The use of two distinct types of data (quantitative and qualitative) had shown a complete picture of the status of PA, its factors, barriers, facilitators and its perceived benefits and harms. But as limitations the cross-section nature of the data cannot show a causal relationship. Besides, the self-report of PA might bias the estimates of PA because of possible memory lapse over a week period.

## Conclusions

The PA of adolescents were generally low among high school students. Student's self-perception about being physically active, and being a member of a sport or fitness team positively associated with their PA along with other variables, namely, perceptions towards their general health as good, and a perception of their family as active, being older, a boy, higher level of vegetable and fruit consumption, the presence of someone who encouraged PA, and perceiving one's family as being active. Having a self-perception of not being overweight, and being in a school that had a sports/playground were negatively associated with their PA. The qualitative finding also showed a consistent finding with the quantitative finding, which indicated that lack of self-motivation, interest and family restrictions as barriers to their PA. Female discussants particularly focused on physical tiredness as barriers to making any vigorous activities.

## Recommendation

Based on the finding of the present study, the following recommendations were made. PA promotion should be made by incorporating PA into school health programs and strengthening the existing school curriculum. Packages of interventions must be designed aimed positively influencing knowledge, attitude and practice of adolescents themselves, as well as their families and peers who were not enrolled in school.

Community health workers in health centers and primary hospitals, and clinicians at various levels of the health care system should advise or teach adolescents who are out-reached by them or reaching their care, respectively, on the benefits of PA, types of PA regarded as MVPA, the recommended weekly PA for the adolescent, and warns about the dangers of physical inactivity. They should also be role models to school adolescents and youths. Awareness creation and training should be provided to students and the school community in a sustainable manner.

## Supporting information

**S1 Raw data. The raw data supporting the finding.**
(XLSX)

## Acknowledgments

We would like to thanks Research Co-ordination Office in college of Medicine and Health sciences, Arba Minch University for providing the opportunity to conduct this study.

## Author Contributions

**Conceptualization:** Eshetu Andarge Zeleke, Teshale Fikadu, Muluken Bekele, Negussie Boti Sidamo.

**Data curation:** Eshetu Andarge Zeleke, Teshale Fikadu, Muluken Bekele, Kidus Temesgen Worsa.

**Formal analysis:** Eshetu Andarge Zeleke, Teshale Fikadu, Muluken Bekele, Negussie Boti Sidamo.

**Investigation:** Eshetu Andarge Zeleke, Muluken Bekele, Kidus Temesgen Worsa.

**Methodology:** Eshetu Andarge Zeleke, Teshale Fikadu, Muluken Bekele.

**Software:** Teshale Fikadu.

**Supervision:** Teshale Fikadu.

**Validation:** Eshetu Andarge Zeleke, Teshale Fikadu, Kidus Temesgen Worsa.

**Visualization:** Eshetu Andarge Zeleke, Teshale Fikadu.

**Writing – original draft:** Eshetu Andarge Zeleke, Teshale Fikadu, Muluken Bekele, Negussie Boti Sidamo, Kidus Temesgen Worsa.

**Writing – review & editing:** Eshetu Andarge Zeleke, Negussie Boti Sidamo, Kidus Temesgen Worsa.

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
