## [Decision Letter · Decision Letter 0]

1 Aug 2023

PONE-D-23-06887Physical activity status among adolescents in Southern Ethiopia: Multi-centered mixed method studyPLOS ONE

Dear Dr. Temesgen,

Thank you for submitting your manuscript to PLOS ONE. After careful consideration, we feel that it has merit but does not fully meet PLOS ONE’s publication criteria as it currently stands. Therefore, we invite you to submit a revised version of the manuscript that addresses the points raised during the review process.

ACADEMIC EDITOR: Dear Author,Please revise your manuscript based on the comments provided by the reviewers. The decision of this manuscript is justified based on PLOS ONE’s publication criteria and not on its novelty or perceived impact.

We look forward to receiving your revised manuscript.

Kind regards,

Zulkarnain Jaafar

Academic Editor

PLOS ONE

Journal Requirements:

2. PLOS requires an ORCID iD for the corresponding author in Editorial Manager on papers submitted after December 6th, 2016. Please ensure that you have an ORCID iD and that it is validated in Editorial Manager. To do this, go to ‘Update my Information’ (in the upper left-hand corner of the main menu), and click on the Fetch/Validate link next to the ORCID field. This will take you to the ORCID site and allow you to create a new iD or authenticate a pre-existing iD in Editorial Manager. Please see the following video for instructions on linking an ORCID iD to your Editorial Manager account: https://www.youtube.com/watch?

Reviewers' comments:

Reviewer's Responses to Questions

**Comments to the Author**

1. Is the manuscript technically sound, and do the data support the conclusions?

Reviewer #1: Yes

Reviewer #2: Yes

2. Has the statistical analysis been performed appropriately and rigorously? 

Reviewer #1: Yes

Reviewer #2: Yes

3. Have the authors made all data underlying the findings in their manuscript fully available?

Reviewer #1: Yes

Reviewer #2: Yes

4. Is the manuscript presented in an intelligible fashion and written in standard English?

Reviewer #1: No

Reviewer #2: Yes

5. Review Comments to the Author

Reviewer #1: Firstly, I would like to commend the authors for their valuable contribution to the field. The study addresses an important research question and presents novel findings that could significantly impact the existing knowledge in the area. The methodology appears to be rigorous and well-designed, and the results are presented clearly.

However, I have a few concerns that I believe need to be addressed before considering the manuscript for publication. These concerns are presented below:

In the introduction section, the authors provided similar information repeatedly, please provide each concept in concise AND ONCE. Similarly, the authors didn't mention the factors and barriers associated with PA, please provide these factors in the introduction section.

Regarding the methods, based on on the information given in the background section, the authors emphasised the importance of physical environment for PA, as a result, its assumed that the authors would have stratified the school children based on the school area (towns), while they simply focused on gender as a source of variation. Any clarification?.

The exclusion criteria are not clearly stated. Moreover, what's the impact of individuals who do not provide consent on the recruitment process and how they will be handled?

Regarding the sampling procedure, the authors selected three out of FIVE government high schools RATHER than two OR four might be, justify it?

Please provide a flow diagram, containing the the study population by town, school and gender.

Overall, the manuscript will benefit from proper proofreading and editing because I cannot list all the language and punctuation mistakes in the text.

Good luck with the revisions and resubmission.

Reviewer #2: Thank you for conducting your study on physical activity, an important public health intervention for the prevention of chronic illness.

General and specific comments are listed below.

General comments

The study is well justified.

The method including the statistical analysis is appropriate and use of mixed method is commendable.

The results are properly presented.

There are editorial issues that need to be addressed.

Dependent variable measurement and quantitative data collection needs further explanation.

Specific comments

The use of space between the end of the sentence and a reference must be carefully corrected throughout the manuscript.

The last three lines on the sample size determination needs correction.

The use of multi-centered is not appropriate as long as the data from the two cities and high schools is not treated in the analysis. I.e., the effect of being an adolescent from high schools in the two cities is not included the multiple regression analyses.

How is the actual data collected conducted? If it was a self-administered, why is the facilitators read each question being at the stage of the classroom?

Physical education is part of their school curriculum which involves different level of physical activity on weekly basis. Is the PA activity that adolescents perform while at school considered in the PA measurement? Depending on the number of Physical education session, students may involve in 50 to 100 minutes.

Instead of removing the three variables from regression because of non-linearity, why not you transformed PA.

6. PLOS authors have the option to publish the peer review history of their article (what does this mean?). If published, this will include your full peer review and any attached files.

Reviewer #1: No

Reviewer #2: No

---

## [Author Response · Author response to Decision Letter 0]

28 Sep 2023

We have revised the manuscript as per your comments

---

## [Decision Letter · Decision Letter 1]

19 Oct 2023

Physical activity status among adolescents in Southern Ethiopia: A mixed methods study

PONE-D-23-06887R1

Dear Dr. Temesgen,

We’re pleased to inform you that your manuscript has been judged scientifically suitable for publication and will be formally accepted for publication once it meets all outstanding technical requirements.

Kind regards,

Zulkarnain Jaafar

Academic Editor

PLOS ONE

Additional Editor Comments (optional):

Reviewers' comments:

Reviewer's Responses to Questions

**Comments to the Author**

1. If the authors have adequately addressed your comments raised in a previous round of review and you feel that this manuscript is now acceptable for publication, you may indicate that here to bypass the “Comments to the Author” section, enter your conflict of interest statement in the “Confidential to Editor” section, and submit your "Accept" recommendation.

Reviewer #1: All comments have been addressed

Reviewer #2: All comments have been addressed

2. Is the manuscript technically sound, and do the data support the conclusions?

Reviewer #1: Yes

Reviewer #2: Yes

3. Has the statistical analysis been performed appropriately and rigorously? 

Reviewer #1: Yes

Reviewer #2: Yes

4. Have the authors made all data underlying the findings in their manuscript fully available?

Reviewer #1: Yes

Reviewer #2: Yes

5. Is the manuscript presented in an intelligible fashion and written in standard English?

Reviewer #1: Yes

Reviewer #2: Yes

6. Review Comments to the Author

Reviewer #1: (No Response)

Reviewer #2: The main concern I had was on the was physical activity status was measured given that it is part of the school curriculum. The authors have now addressed it. Therefore, the revision is acceptable.

7. PLOS authors have the option to publish the peer review history of their article (what does this mean?). If published, this will include your full peer review and any attached files.

Reviewer #1: No

Reviewer #2: No

---

## [Editor Report · Acceptance letter]

27 Oct 2023

PONE-D-23-06887R1 

Physical activity status among adolescents in Southern Ethiopia: A mixed methods study 

Dear Dr. Temesgen Worsa:

I'm pleased to inform you that your manuscript has been deemed suitable for publication in PLOS ONE. Congratulations! Your manuscript is now with our production department. 

Kind regards, 

on behalf of

Dr. Zulkarnain Jaafar 

Academic Editor

PLOS ONE